# The Population Dynamics and Parasitism Rates of *Ceratitis capitata*, *Anastrepha fraterculus*, and *Drosophila suzukii* in Non-Crop Hosts: Implications for the Management of Pest Fruit Flies

**DOI:** 10.3390/insects15010061

**Published:** 2024-01-15

**Authors:** María Josefina Buonocore-Biancheri, Xingeng Wang, Segundo Ricardo Núñez-Campero, Lorena Suárez, Pablo Schliserman, Marcos Darío Ponssa, Daniel Santiago Kirschbaum, Flávio Roberto Mello Garcia, Sergio Marcelo Ovruski

**Affiliations:** 1Planta Piloto de Procesos Industriales Microbiológicos y Biotecnología (PROIMI-CONICET), División Control Biológico, Avda. Belgrano y Pje. Caseros, San Miguel de Tucumán T4001MVB, Argentina; mjbuonocoreb@hotmail.com (M.J.B.-B.); mdpwolf@gmail.com (M.D.P.); sovruski@conicet.gov.ar (S.M.O.); 2USDA-ARS Beneficial Insects Introduction Research Unit, Newark, DE 19713, USA; 3Centro Regional de Investigaciones Científicas y Transferencia Tecnológica de La Rioja (CRILAR-CONICET), Entre Ríos y Mendoza s/n, Anillaco, La Rioja 5301, Argentina; segundo.nc@conicet.gov.ar; 4Departamento de Ciencias Exactas, Físicas y Naturales, Instituto de Biología de la Conservación y Paleobiología, Universidad Nacional de La Rioja (UNLaR), Av. Luis M. de la Fuente s/n., La Rioja 5300, Argentina; 5Dirección de Sanidad Vegetal, Animal y Alimentos de San Juan (DSVAA)-Gobierno de la Provincia de San Juan, CONICET, Nazario Benavides 8000 Oeste, Rivadavia, San Juan J5413ZAD, Argentina; lorenasuarez@conicet.gov.ar; 6CCT CONICET San Juan, Argentina Av. Libertador Gral. San Martín 1109, San Juan J5400AR, Argentina; 7Centro Regional De Energía y Ambiente para el Desarrollo Sustentable (CREAS), CONICET-UNCA, Prado 366, de Catamarca 4700 SFV, Argentina; schliserman73@yahoo.com.ar; 8INTA Estación Experimental Agropecuaria Famaillá, Tucumán Ruta Prov. 301, km 32, Famaillá 4132, Argentina; kirschbaum.daniel@inta.gob.ar; 9Cátedra Horticultura, Facultad de Agronomía y Zootecnia, Universidad Nacional de Tucumán, San Miguel de Tucumán 4000, Argentina; 10Departamento de Ecologia, Zoologia e Genética, Instituto de Biologia, Universidade Federal de Pelotas, Pelotas 96000, Rio Grande do Sul, Brazil; flavio.garcia@ufpel.edu.br

**Keywords:** medfly, spotted-wing drosophila, South American fruit fly, seasonal infestation level, fruit fly abundance, parasitoid, non-crop host, disturbed natural habitat

## Abstract

**Simple Summary:**

Non-crop host plants inhabiting wild vegetation areas surrounding crops strongly influence the dynamics and abundance of polyphagous pest fruit flies, including *Ceratitis capitata* (*Cc*), *Drosophila suzukii* (*Ds*), and *Anastrepha fraterculus* (*Af*). The two former species are dangerous invasive pests widespread in all Argentinean fruit-producing regions, whereas the latter species, native to the Neotropics, coexists with those exotic species in northwestern Argentina. Integrated and eco-friendly management strategies are needed against those pests, targeting both crop and non-crop areas. Therefore, this study assessed the abundance of these pest dipterans, their seasonal infestation levels in five non-crop fruit species, relationships with competing saprophytic drosophilids, and natural parasitism. Fruits were surveyed in a disturbed wild habitat in northwestern Argentina over 40 months, and fruits were sampled from the tree canopies and ground. The results revealed that *Af* had the highest abundance, followed by *Cc* and *Ds*. Saprophytic drosophilids were predominant only from ground fruit samples. Spatiotemporal overlaps of different host fruit availability enabled continuous and suitable sources for pest proliferation throughout the year. The population peaks of both exotic pests coincided with the highest availability of peaches from December to January, whereas the *Af* population peaked during guava fruiting from February to April. These pest flies were attacked mainly by generalist parasitoids that could be useful in the conservation and augmentative biological control of these pests.

**Abstract:**

Understanding the seasonal dynamics inherent to non-crop host–fruit fly–parasitoid interactions is vitally important for implementing eco-friendly pest control strategies. This study assessed the abundance and seasonal infestation levels of three pest fly species, *Ceratitis capitata* (Wiedemann), *Anastrepha fraterculus* (Wiedemann), *Drosophila suzukii* (Matsumura), as well as the related saprophytic drosophilids, and their natural parasitism in a disturbed wild habitat characterized by non-crop hosts in northwestern Argentina over 40 months. *Juglans australis* Griseb (walnut), *Citrus aurantium* L. (sour orange), *Eriobotrya japonica* (Thunb.) Lindley (loquat), *Prunus persica* (L.) Batsch (peach), and *Psydium guajava* L. (guava) were sampled throughout their fruiting seasons. Fruits were collected from both the tree canopies and the ground. The most abundant puparia was *A. fraterculus*, followed by *C. capitata* and *D. suzukii*. *Drosophila* species from the *D. melanogaster* group were highly abundant only in fallen fruits. Spatiotemporal overlaps of different host fruit availability provided suitable sources for pest proliferation throughout the year. The populations of both invasive pests peaked from December to January, and were related to the highest ripe peach availability, whereas the *A. fraterculus* population peaked from February to April, overlapping with the guava fruiting period. The three pest fly species were parasitized mainly by three generalist resident parasitoids, which are potential biocontrol agents to use within an integrated pest management approach.

## 1. Introduction

Landscape fragmentation plays an essential role in the establishment, dispersal, and population dynamics of invasive species in a new location [1]. Disturbance of the natural habitat strongly influences the composition and abundance of related biota [2,3,4]. This occurs through the competitive displacement of native species, changes in natural enemy abundance and diversity, and the capacity of the invader to occupy empty or disturbed niches, among other factors [5]. In the case of invasive fruit flies, the distribution and abundance of host plants, the structure of vegetation surrounding crops as alternative habitats, and the distribution of essential resources such as food, shelter, and oviposition substrates strongly influence the spatiotemporal dynamics, distributions, and abundances of fruit fly pests [6,7,8]. Representative examples of habitat-driven pest dynamics are global invasive species *Ceratitis capitata* (Wiedemann) (Diptera: Tephritidae), native to Mediterranean Africa and commonly known as medfly [9], and *Drosophila suzukii* (Matsumura) (Diptera: Drosophilidae), originally from Southeast Asia and known worldwide as the spotted-wing Drosophila [10,11]. These two exotic fruit fly species are severe pests of economically valuable fruit crops worldwide [12,13], although *D. suzukii* mainly attacks soft-skinned small fruits, such as berries and cherries [14]. Unlike most *Drosophila* species, *D. suzukii* females lay eggs in fresh, healthy, ripening fruit because it has a serrated ovipositor, which allows females to oviposit inside the fleshy mesocarp [15]. Another relevant example of the habitat-driven pest population dynamic is the Neotropical-native *Anastrepha fraterculus* (Wiedemann) (Diptera: Tephritidae). This tephritid fruit fly is the most economically important species of *Anastrepha* in South America, and it is a quarantine pest for the United States and several European and Asian countries [16]. All three dipteran species are highly polyphagous and can exploit various crop and non-crop host plants [9,17,18]. 

The availability of alternative hosts in non-crop habitats could play an important role in sustaining the populations of polyphagous fruit flies and dictating their local movement patterns when favorable hosts are not available in crops, and non-crop habitats could act as sinks, sources, shelters, or overwintering sites for the fly populations [7,19,20,21]. Thus, the effectiveness of any control measures for those polyphagous and highly mobile pests requires in-depth knowledge of their seasonal field ecology, including the role of non-crop host plants in the landscape structure for population dispersal and persistence [22]. Such information is essential to implement integrated and area-wide pest management strategies that minimize environmental impact and maximize sustainability to reduce reliance on insecticides alone [23,24,25,26,27]. In this context, resident natural enemies may play a unique role in reducing insect pest populations in non-crop environments that could provide reservoirs for the pest populations moving into crops after they have been treated [28,29,30]. Non-crop hosts can also provide various ecological services to neighboring agricultural environments, including maintaining and amplifying the numbers of beneficial insects, such as parasitoids [31]. Therefore, biological control properly used in natural environments may be a valuable option for long-term, landscape-level management of insect pests [32,33,34,35]. 

The subtropical mountain rainforest, locally known as Yungas, is one of the South American mountain cloud forests divided into sections along an altitudinal gradient that extends discontinuously from Venezuela to northwestern Argentina [36]. In Argentina, the Yungas lowlands have been strongly transformed into crop and pasture areas because of agricultural development and human settlement [36]. However, in the last five decades, vast sectors of croplands have been restored as nature conservation areas. Thus, the native vegetation naturally regenerated, although abundant exotic plants also grew [37]. Some feral plants have been recorded as hosts of *C. capitata*, *A. fraterculus*, and *D. suzukii*, coupled with local parasitoid assemblages [38,39,40,41,42]. Therefore, natural sites with high and medium disturbance levels are interesting frameworks to evaluate how non-crop hosts adjacent to fruit crops can increase the risk of infestation during the fruiting season.

The current study aimed to describe the abundances and infestation levels of *C. capitata*, *A. fraterculus, D. suzukii* and related saprophytic drosophilids infesting five prevalent non-crop fruit species in a highly disturbed natural habitat adjacent to commercial crops and family orchards in northwestern Argentina. Although most saprophytic drosophilids are not considered pests, they share many generalist drosophila parasitoids with *D. suzukii* and may act as alternative hosts for these parasitoids. We compared temporal variations of the infestation levels by the three pest dipteran species during the fruiting seasons of the five host fruit species and assessed natural parasitism levels. Focusing on the tri-trophic interaction (host fruit–fruit fly–parasitoid) over a long-term period in a disturbed wild area would allow a better understanding of how the three fruit fly pests use non-crop fruits based on temporal patterns of host availability. Simultaneously, it is also feasible to identify key hosts accountable for pest population increase, persistence, and the incidence of resident parasitoids in the landscape as the season progresses. This information is useful for not only the different fruit-growing regions of Argentina but also for regions of Latin America and throughout the world affected by some of those pest dipterans.

## 2. Materials and Methods

### 2.1. Study Area

The area, located in Horco Molle, Yerba Buena district, Tucumán province, northwestern Argentina, originally belonged to the Low Montane Forest sector from the southernmost end of the subtropical mountain Yungas Forest [36]. The study site belongs to the Horco Molle Experimental Reserve (HMER), a protected wildlife area. This area lies between 26°47′ S latitude and 65°18′ W longitude at 600 m and covers a total surface area of 200 ha. Adjacent to the HMER is the Sierra de San Javier Park, both managed by the National University of Tucumán. A disturbed secondary rainforest, i.e., both exotic and native plant species co-exist, characterizes the study site (Photograph, Appendix A). The surrounding landscape is a mosaic of various commercial citrus crops, small familiar multi-fruit orchards, scattered rural houses, and wild secondary forest patches, with the closest crops located < 0.5 km away from the study site (Scheme, Appendix A). The climate in this region is classified as “humid warm–temperate” with a rainy warm season from October through April, and a dry cold season from May through September, with ≈22 °C and 900 mm of average annual temperature and rainfall, respectively [43]. The variation in mean temperature and accumulated rainfall during collecting periods at the study site is shown in Figure 1A.

### 2.2. Host Fruit Sampling

A total of 56, 176, 54, 64, and 72 *Juglans australis* Griseb (wild walnut) (Juglandaceae), *Citrus aurantium* L. (sour orange) (Rutaceae), *Eriobotrya japonica* (Thunb.) Lindley (loquat) (Rosaceae), *Prunus persica* (L.) Batsch (peach) (Rosaceae), and *Psydium guajava* L. (guava) (Myrtaceae) trees, respectively, were sampled according to the temporal patterns of fruit ripening throughout their fruiting seasons (Figure 1B). Two trees of each species were chosen randomly on bi-weekly sampling dates from November 2016 to March 2020. Plants were not sampled after March 2020 due to the confinement established by the Argentinian government due to the COVID-19 pandemic. These five non-crop hosts are highly abundant and widely spread throughout disturbed wildland areas of northwestern Argentina [44]. *Juglans australis* was the only native species sampled, and the remainder were feral exotic species. Sample size varied according to fruit weight and relative fruit availability per host species. Fruit samples by the collection date were 48, 16, 200, 60, and 40 wild walnuts, sour oranges, loquats, peaches, and guavas, respectively. Half of the ripe fruit in each sample was randomly collected from the tree canopies and the remaining half from the ground beneath the canopies. Fruits were separately handled to determine whether there were differences in both fly and parasitoid species composition at each level. To collect fruit located in canopies above 1.8 m high, a plastic basket attached to a 3.5 m long extendable metal pole was placed beneath the fruit, and the branch was shaken. Each fruit sample was placed individually into a 20 × 30 m (diameter × deep) cloth bag and transported in plastic crates for processing at the Pest Biological Control Department (DCBP, Spanish acronym). This department belongs to the Biotechnology and Microbiological Industrial Processes Pilot Plant (PROIMI, Spanish acronym) in San Miguel de Tucumán, Tucumán, 15 km from the study site.

### 2.3. Host Fruit Processing

All collected fruits from the canopies or ground were rinsed with a 30% sodium benzoate and 70% sterile water solution and weighed individually. Mean (±SE) individual fruit weight was 39.7 ± 1.1, 126.4 ± 3.5, 10.5 ± 1.0, 34.5 ± 2.3, and 49.5 ± 1.7 g for walnut, sour orange, loquat, peach, and guava, respectively. Fruit from the ground and the canopies were separately processed and kept individually. First, each fruit was placed in a 48 × 28 × 15 cm plastic crate with a slotted bottom. Then, the crate was placed over another plastic crate of the same size but with a non-perforated bottom and with a thin layer of sterilized, moistened vermiculite Intersum^®^ (Aislater S.R.L., Cordoba, Argentina) on the bottom as a pupation medium. Both crates were tightly covered with a shiny polyester organza fabric lid. The double crate method prevented mixing sand with fruit, fungal growth, and bacterial contamination. All collected samples from the same date were grouped on shelves, which were kept in a dark room under natural environmental conditions for two weeks. Vermiculite was sifted daily to collect fly puparia. Finally, each fruit was dissected to search for larvae or puparia remaining inside the fruit.

### 2.4. Fly Puparia Processing and Identification

Fly puparia were identified at the DCBP’s laboratory. Both *A. fraterculus* and *C. capitata* puparia were identified using external characters of everted anterior spiracles, tubes with finger-like projections [45]. *Drosophila suzukii* puparia were also differentiated from those saprophytic drosophilids by the external characteristic shape of the anterior spiracles [46]. Puparia of different saprophytic drosophilid species were not identified. The puparia of each fly species belonging to the same fruit sample were processed separately. Then, they were transferred to 200 cc translucent plastic cups filled with sterilized moist vermiculite. Each cup was covered with a shiny polyester organza fabric and tied with a rubber band. Cups were placed into 32 × 24 × 12 cm plastic containers. Each container housed the puparia of a particular fly species from the same fruit sample. The numbers of emerged flies and parasitoids were recorded weekly. Voucher adult specimens were stored at the entomological collection of the Fundación Miguel Lillo in San Miguel de Tucumán.

### 2.5. Data Analysis

The response variables analyzed were the monthly accumulated fruit infestation level by fly species, infestation level recorded in each fruit species by fly species, total parasitism on each fly species, and the parasitoid abundance per fly species. All variables were estimated for both fallen and canopy fruit samples. The fruit infestation level was calculated as the total number of recovered fly puparia per 100 g of fruit weight. The monthly accumulated infestation level was calculated by combining infestation values obtained from all host fruit species during a particular month and by fly species. The infestation level recorded by host fruit species was calculated by including infestation values recorded over a 40-month survey period and by fly species. The total parasitism on each fly species was calculated as the total adult parasitoid number over the total number of puparia recovered from a particular fly species throughout all collecting periods, regardless of host fruit species. The parasitoid abundance was calculated as the total number of parasitized puparia by host fly species from all fruit species collected over the 40-month survey period. The statistical analysis was performed using the software R-4.3.2 [47]. Kruskal–Wallis’ rank sum tests were performed to compare fruit infestation levels and parasitoid abundance per fly species. Dunn’s post hoc pairwise comparison tests were conducted to show differences between factor levels using a Bonferroni–Holm adjustment method. Mann–Whitney–Wilcoxon tests, with a Bonferroni–Holm adjustment method, were performed to compare parasitism on fly species recovered from both canopies and ground fruit samples. Violin box plots were used to show the resulting data. Violin box plots were used for the figures with statistical data. A violin plot is a mixture of a box plot and a kernel density plot, which shows peaks in the data. Figures were made with the ‘grouped_ggbetweenstats’ function from the ‘ggstatsplot’package [48]. Each plot involves media (horizontal line inside the box), median (red dot inside the box), interquartile range Q1–Q3 (vertical line inside the box), range (minimum: Q0, maximum: Q4; both ends of the whisker on the vertical line outside the box), and raw data dispersal (colored circles). The library ‘rcompanion’ function was used to include letters that display the significant difference in figures.

## 3. Results

### 3.1. Fly Abundance and Infestation Levels

A total of 11,212 fruits (408.8 kg) were collected, 50% from the tree canopies and 50% from the ground during this study, which yielded 19,989 *A. fraterculus*, 19,187 *C. capitata*, 3242 *D. suzukii*, and 23,999 *Drosophila* spp. puparia (Table 1). Saprophytic *Drosophila* species were from the *Drosophila melanogaster* species group. Tephritid puparia accounted for 59% of the total recovered fly puparia, whereas the remaining 41% were drosophilid puparia, from which only 12% belonged to *D. suzukii*. Fruit infestation levels by the three pest dipteran species varied sharply across sampling months (Figure 1C). *Ceratitis capitata* yielded significantly the highest infestation levels particularly between November and February, with a peak in January, in fruits collected either from canopies (χ^2^_kruskal-Wallis (11)_ = 125.75, *p* < 0.0001) (Figure 2A) or from ground (χ^2^_kruskal-Wallis (11)_ = 109.75, *p* < 0.0001) (Figure 2B). 

*Anastrepha fraterculus* showed significantly the highest infestation levels between December and May in fruits sampled either from canopies (χ^2^_kruskal-Wallis (11)_ = 85.08, *p* < 0.0001) (Figure 2C) or from the ground (χ^2^_kruskal-Wallis (11)_ = 130.00, *p* < 0.0001) (Figure 2D). *Drosophila suzukii* exhibited the highest infestation levels between October and May, although infestation peaked between November and January in fruits collected either from canopies (χ^2^_kruskal-Wallis (11)_ = 29.59, *p* < 0.0001) (Figure 2E) or from the ground (χ^2^_kruskal-Wallis (11)_ = 49.58, *p* < 0.0001) (Figure 2F). Saprophytic drosophilids had significantly similar infestation levels, <1 fly puparium/100 g fruit, in fruits collected from canopies throughout the year (χ^2^_kruskal-Wallis (11)_ = 23.31, *p* = 0.0160) (Figure 2G). Infestation levels by saprophytic drosophilids were remarkably high in fallen fruits from January to April (χ^2^_kruskal-Wallis (11)_ = 200.97, *p* = 0.0001) (Figure 2H). Infestation levels by the three pest dipteran species and by saprophytic *Drosophila* species showed significant differences among the different fruit species, collected either from the canopies or from the ground (Table 2). Significantly higher infestation levels by *A. fraterculus* than those of the other pest fly species were recorded from walnut (Figure 3A,B) and guava (Figure 3E,F), whereas *C. capitata* had significantly the highest infestation levels in peach (Figure 3I,J), loquat (Figure 3C,D), and sour orange (Figure 3G,H). Infestation levels by *D. suzukii* in peach were high, but similar to that of *A. fraterculus* (Figure 3I,J). Infestation levels by *Drosophila* spp. from *D. melanogaster* group were the highest in all sampled fruit species, but only in fruit samples collected from the ground (Figure 3B,D,F,H,J).

### 3.2. Parasitoid Abundance and Parasitism Levels

A total of 7349 adult parasitoids belonging to six different species, *Ganaspis pelleranoi* (Brèthes) (28.6%) (Figitidae), *Trichopria anastrephae* Lima (28.2%) (Diapriidae), *Pachycrepoideus vindemiae* Rondani (18.1%) (Pteromalidae), *Leptopilina* sp. cf. *boulardi* (Barbotin, Carton, and Kelner-Pillault) (Figitidae) (14.9%), *Doryctobracon areolatus* (Szèpligeti) (5.7%) (Braconidae), and *Doryctobracon brasiliensis* (Szèpligeti) (4.5%) (Braconidae), were obtained from fly puparia recovered over the 40-month study. Five parasitoid species, *D. areolatus*, *D. brasiliensis*, *G. pelleranoi*, *P. vindemiae*, and *T. anastrephae*, were recovered from *A. fraterculus*, whereas only *G. pelleranoi* and *P. vindemiae* were associated with *C. capitata*, and *T. anastrephae*, *Leptopilina* sp. cf. *boulardi*, and *P. vindemiae* with both *D. suzukii* and *Drosophila* spp. The latter three parasitoid species prevailed on saprophytic drosophilids, whereas *G. pelleranoi* mostly parasitized *A. fraterculus* and to a minor extent *C. capitata* (Figure 4). The braconid species were found as associated only with *A. fraterculus* (Figure 4). The numbers of parasitized host puparia recorded in all three pest fly species and in saprophytic drosophilid species were significantly different between the ground (χ^2^_kruskal-Wallis (3)_ = 1298.81, *p* < 0.0001) (Figure 5A) and canopy (χ^2^_kruskal-Wallis (3)_ = 281.66, *p* < 0.0001) (Figure 5B) fruit samples. The highest number of parasitized host puparia was on saprophytic drosophilids recovered from fallen fruits (Figure 5A). The number of parasitized *C. capitata* puparia was significantly higher than that of *A. fraterculus* and both were significantly higher than that of *D. suzukii* (Figure 5A). The number of parasitized *A. fraterculus* puparia recorded from the canopy fruit was significantly higher than that recorded for other tested fly species (Figure 5B). Moreover, the number of parasitized *C. capitata* puparia was significantly higher than that recorded from both *D. suzukii* and saprophytic drosophilids (Figure 5B). Significant positive correlations between parasitism and infestation levels were recorded for *C. capitata* (τ = 0.51, z = 18.75, *p* < 0.0001), *A. fraterculus* (τ = 0.75, z = 27.10, *p* < 0.0001), *D. suzukii* (τ = 0.37, z = 11.85, *p* < 0.0001), and *Drosophila* spp. (τ = 0.85, z = 30.34, *p* < 0.0001). The total levels of parasitism were significantly different between the host puparia recovered from fruits still in the canopies and those from fallen fruits. Significantly, more parasitoids were recovered from puparia collected from fallen fruits than from the canopy fruits. This pattern was consistent for *A. fraterculus* (*W_M-W_* = 7.01^5^, *n* = 2450, *p* < 0.0001) (Figure 6A), *C. capitata* (*W_M-W_*= 1.59^5^, *n* = 1384, *p* < 0.0001) (Figure 6B), *Drosophila* spp. (*W_M-W_*= 7650, *n* = 1221, *p* < 0.0001) (Figure 6C), and *D. suzukii* (*W_M-W_* = 1.01^5^, *n* = 975, *p* < 0.0001) (Figure 6D). 

## 4. Discussion

The current study provides significant information needed to develop fruit fly IPM strategies that minimize environmental impact and maximize long-term sustainability. The results showed that (1) highly disturbed wild habitats adjacent to crops are suitable sites for the development and increase of the pest fruit fly species *C. capitata*, *A. fraterculus*, and *D. suzukii*; (2) non-crop host fruit species influence the relative and temporal abundances of these flies; (3) overlaps in fruiting seasons of different host species throughout the year allow these flies to access regularly resources to sustain their populations in the disturbed habitats; (4) temporal infestation levels by both invasive pest species are similar but differ from the native pest; and (5) the abundance and diversity of resident parasitoids, as well as parasitism levels, depend largely on non-crop fruit species where the larval or pupal hosts developed, dipteran host species associated with the host fruits, and fruit infestation levels.

Firstly, we found high abundance and fruit infestation levels by the three fly species in a forest regenerated from anthropogenic disturbances. Many characteristics in the disturbed habitats, such as the presence of abundant and diverse indigenous and exotic host fruit species as well as the high thermal and humidity variation may allow the occurrence and coexistence of these species. Some introduced host plants widely dispersed in this habitat such as *C. aurantium* and *E. japonica*, are uncontested or poorly contested by native fly species, thereby providing empty niches mainly exploited by *C. capitata* [38], but less so by *D. suzukii*. In addition to the high level of polyphagy of *C. capitata* and *D. suzukii*, both exotic flies have high thermal plasticity, allowing them not only to persist but also to thrive in such disturbed environments [6,9,49,50,51,52]. Although *A. fraterculus* prevails in low-disturbed environments with a high abundance of native plants, it is also usually found in association with exotic host fruits in highly disturbed environments [38,53].

Secondly, this study revealed different preferences for certain host plants among the three fly species. *Prunus persica* was the preferred host fruit for *C. capitata*, followed in decreasing order by *P. guajava*, *E. japonica*, *C. aurantium*, and *J. australis*. It was evident that *C. capitata* preferred introduced feral fruits that are usually underutilized by *A. fraterculus*, with the exception of *P. guajava*. Previous records [38,39] pointed out feral *P. persica* as the most relevant multiplying hosts in northern Argentina and one of the key hosts for *C. capitata* dispersing in all Argentinian fruit-growing regions. Feral *P. guajava* was also the preferred exotic host for *A. fraterculus* and together with the native *J. australis* were the ones that mainly allowed the highest population growth of *A. fraterculus* as shown in this study. Interestingly, the presence of *P. guajava* in disturbed habitats characterized by low native plant cover and by sectors with a higher incidence of sun, increasing *A. fraterculus*’s abundance, although preferred native hosts, such as walnuts, are still present. This occurs because *P. guajava* is the most commonly recorded *A. fraterculus* host plant throughout the Neotropics [17]. Interestingly, *J. australis* had not previously been recorded as a host of *D. suzukii* in Argentina or South America. Although *D. suzukii* was occasionally abundant in that native fruit species, it is a novel host to 24 exotic and native, crop, and non-crop fruits thus far recorded for this invasive pest in Argentina [40]. *Drosophila suzukii* was preferentially more abundant in *P. persica* in the study site, followed in decreasing order by *P. guajava* and *E. japonica*. Both feral peach and guava have previously been recorded as alternative hosts to *D. suzukii* in wilderness areas in northwestern Argentina [40]. Similarly, *D. suzukii* was previously recorded infesting *E. japonica* fruits in crop areas of northwestern Argentina [54], as well as in commercial peaches in northeastern Buenos Aires (central-eastern Argentina) [55]. Data from the current study on the abundance of *D. suzukii* on *P. persica* and *E. japonica* are not surprising since Rosaceae is the plant family with the largest number of host species recorded for *D. suzukii* worldwide [21]. *Citrus aurantium* evidently is not a suitable oviposition host for *D. suzukii*. However, two *Citrus* species, *C. sinesis* (L.) Osbeck and *C. reticulata* Blanco, have been recorded as alternative reproductive hosts on damaged fruits in California (USA) [56], whereas *C. sinensis* was also recorded in Uruguay [21]. Rutaceae apparently include host species not preferred by *D. suzukii* [21]. The high abundance and high infestation levels of saprophytic drosophilids on all sampled fruit species can be mainly attributed to the fact that these dipterans are associated with a wide variety of habitats, particularly related to rotting fallen fruits [57]. Precisely, data from the current study show the highest infestation levels of these drosophilids in ripe fruits sampled only from the ground.

Thirdly, we showed overlaps of temporal availability of *P. persica* with the remaining host fruit species and a constant availability of ripe *C. aurantium* fruit throughout the year. This provides these fruit fly pest species with year-round resources for oviposition in the study site. In this context, mainly both *E. japonica* and *C. aurantium*, but also *P. guajava*, play important bridging roles during the cold-dry season, which spans from late autumn and winter to early spring. During this period of the year, *P. persica* is not available and its availability is low throughout mid- and late autumn when compared to the summer and early autumn seasons. The role of *E. japonica* as a host for the three fly pests is crucial, despite the low infestation levels recorded for this exotic, feral fruit species. This is because *E. japonica* provides an alternative host when the latest guavas are not highly available until late autumn, and when the earliest peaches ripe in late spring. This was previously recorded only for *C. capitata* and *A. fraterculus* [38]. This study also showed that *D. suzukii* used the same resource as the other two tephritid fly pests to persist at low density during a period of unfavorable climatic conditions and a shortage of primary hosts. This is new information on the ecological aspects of *D. suzukii* in northwestern Argentina, as the loquat apparently is also a reservoir host for this invasive pest, whereas *C. aurantium* is a non-host. This fact is relevant because *P. persica* is not only the main multiplying host for *C. capitata*, but also for *D. suzukii*, as *P. guajava* is for *A. fraterculus* and the second proliferating host for both *C. capitata* and *D. suzukii*. Therefore, *D. suzukii* may be found throughout the year in environments with floristic characteristics similar to the site of the current study. This is mainly due to the presence of late ripe guavas and the early ripening of the loquat during the dry cold period. Isolated *D. suzukii* adult catches in liquid traps were recorded in August (mid-winter) in blueberry-growing lowland areas of Tucumán [58]. The presence of infested feral loquats in wild vegetation areas surrounding berry crops may explain the winter catches of *D. suzukii* adults. Similarly, in southern Brazil, *D. suzukii* can still remain at low natural infestation rates in native non-crop hosts, such as *Psidium cattleianum* Sabine (strawberry guava) and *Eugenia unifora* L. (surinam cherry), and in feral loquat, even in winter [59]. Therefore, *D. suzukii* females that have overwintered in alternative non-crop host fruits are probably a source of infestations in crop fruits available during spring in northwestern Argentina. *Drosophila suzukii* has high dispersal abilities, which enable it to move freely between both non-crop and crop habitats throughout the year [22,60,61,62,63]. The same dispersal behavior between crops and patches of wild vegetation and surrounding family gardens in a heterogeneous landscape has been recorded for *C. capitata* [44,64,65] and *A. fraterculus* [53,66,67]. Structurally complex landscapes influence trophic interactions mainly because suitable resources occurring in different types of patches can support consumer species [68]. This essentially shapes the spatial and temporal dynamics of the biological communities in these landscapes [69].

Fourthly, we showed that the population dynamics of *D. suzukii* and *C. capitata* appeared to be similar but partially differ from *A. fraterculus*. Both *D. suzukii* and *C. capitata* populations gradually increased from August (cold dry winter), reaching the highest peak in January in *C. capitata*, and between December and January in *D. suzukii* (warm-humid summer) and then sharply declined in March to maintain a low abundance throughout autumn and winter. The population peaks may be associated with the highest availability of peach and, to a lesser degree with the walnut fruiting period. However, the availability of guava may have also influenced the infestation level of *D. suzukii* during February. These low populations are not only associated with the absence of preferred host fruits but also essentially due to the decrease in temperature and humidity at the end of the warm humid season as previously discussed [44,58]. Earlier studies [38] in a secondary forest of northwestern Argentina indicated two continuous population peaks for *C. capitata* in December and January, coinciding with the greatest availability of both *P. persica* and *C. aurantium*. In the current study, only one population peak was detected in January, as accumulated infestation levels recorded for *C. capitata* in December and February were very similar and lower than in January. The native host *J. australis* played a relevant role in increasing *C. capitata* population, as in January the infestation level was 2.7-fold higher than that of *C. aurantium*. As for *D. suzukii*, the current study provides first-hand information on the temporal abundance variation of this invasive pest in Argentina, because the few known studies on population fluctuation of this pest in Argentina were only carried out in berry-growing areas using trap catches of adult flies. In northern and central Argentinian fruit-producing regions, trap catches detected two adult population peaks in late spring–early summer (November and December) and in mid-autumn (April and May), respectively, with the catches being lower in the second than the first peaks and declining from late autumn onward [54,55,58,70,71,72]. However, in the Alto Valle de Rio Negro, northern Patagonia (i.e., in the cold and dry southern Argentina), the peak of trapped *D. suzukii* adults occurred between late summer and late autumn (March-May), coinciding with raspberry and cherry fruiting seasons [73]. Climatic conditions are probably the major factors affecting *D. suzukii* abundance [74]. The hottest and coldest months of the year in temperate and subtropical climates may reduce *D. suzukii* populations; therefore, this pest usually increases its population in late spring and mid-autumn [75]. This indeed reflects the population dynamics based on adult catches in berry-growing areas from Argentina but is not consistent with data of the current study, because the major *D. suzukii* population peak occurred during the month with high temperature and humidity. The diverse microhabitats in this environment and the phenotypic and thermal plasticity of *D. suzukii*, as well as a high availability of suitable fruits, are probably responsible for the population increase in the middle of the warm humid season. Similarly, the infestation levels of *A. fraterculus* also gradually increased from August as *C. capitata* and *D. suzukii*, but the infestation levels of *A. fraterculus* continually increased after January and reached population peaks between February and April. This coincides with the guava fruiting period and the gradually rising temperature and humidity as summer progresses [44]. Infestation levels *A. fraterculus* decreased sharply after May and remained low during late autumn and throughout winter.

Finally, the current study revealed the trophic associations among these host plants, dipteran pests, and resident parasitoid species as well as the relative abundance and diversity of parasitoids throughout the year. Although *C. capitata* was the dominant pest fly in three feral introduced fruit species, C. *aurantium*, *E. japonica,* and *P. persica*, it was parasitized only by *G. pelleranoi* and *P. vindemiae*, both generalist parasitoids [76]. Similarly, *C. capitata* was also only parasitized by these two parasitoids on *P. guajava* and *J. australis*, a major host of *A. fraterculus*. The figitid *G. pelleranoi* is one of the few Neotropical-native larval parasitoid species sympatrically associated with *Anastrepha* that can successfully develop on *C. capitata* larvae [76]. *Ganaspis pelleranoi* females frequently forage fly larvae inside fallen fruit and mainly attack the host by entering through the fissures produced in the fruit or holes produced by its jaws [77]. Faced with this behavior, physical features, such as large size, rind thickness, and pulp depth, do not limit the parasitoid’s access to locate and parasitize host larvae. This was supported in the current study as 85% of the total identified *G. pelleranoi* specimens were from fallen fruit samples. The pteromalid *P. vindemiae* is a cosmopolitan species that attacks puparia of various cyclorrhaphous dipteran species, among which, *C. capitata* is a host recurrently recorded in the American continent [76]. *Pachycrepoideus vindemiae* is an abundant and widespread species in wild vegetation environments from northwestern Argentina, where it was recorded as a common pupal parasitoid on *C. capitata* [39]. In terms of parasitoid diversity and abundance, *C. capitata* was parasitized by two of the six identified species (33%), but the abundance of parasitoids associated with this invasive pest was high. The abundance of *G. pelleranoi* recovered from *C. capitata* prevailed on highly available fruits during the warm humid season, while *P. vindemiae* from *C. capitata* was more abundant (76%) than *G. pelleranoi* only on loquat. This may be because loquat is mostly available during the cool dry season, a time of the year with low *C. capitata* infestation levels, and the absence of *G. pelleranoi* in the study area. Interestingly, *P. vindemiae* was recovered from *C. capitata* puparia collected from loquat from mid-August to mid-November. Apparently, *P. vindemiae* is a parasitoid not only with high adaptability to diverse environments but also with greater thermal plasticity than native parasitoids, such as *G. pelleranoi*. The other invasive species, *D. suzukii*, also showed low parasitoid diversity. Only three species were recovered, with *P. vindemiae* as the prevalent parasitoid in the four host plant species associated with *D. suzukii*, followed by *T. anastrephae*, but only in *P. persica* and *P. guajava*. In line with the latter, both host fruit species had the highest infestation levels by *D. suzukii*. However, the abundance of *P. vindemiae* recovered from *D. suzukii* was low compared with that of the other two identified flies, but *T. anastrephae* was mainly abundant on *D. suzukii*, rare on *A. fraterculus* and absent on *C. capitata.* The South American-native *T. anastrephae* is a pupal endoparasitoid previously associated with both *A. fraterculus* and *D. suzukii* in Argentina [39] and Brazil [54]. However, *T. anastrephae* has a strong preference for parasitizing puparia of resident saprophytic drosophilid species located inside the fruit [42]. The low diversity of parasitoids associated with *D. suzukii* in the study area may correlate with the absence of host–parasitoid co-evolution and co-adaptation processes, especially for larval endoparasitoids that must overcome the hosts’ immune response, and for this reason, they are highly co-evolved with their particular hosts. This also applies to the case of *C. capitata* as correlation coefficients between parasitism and fruit infestation by *D. suzukii* and *C. capitata* were between 1.5- and 2.3-fold lower than those recorded for both *A. fraterculus* and *Drosophila* spp. (*D. melanogaster* group). Although some larval parasitoid species were recovered from *D. suzukii* puparia in Argentina, such as *Dieucoila octofagella* Reche, *Ganaspis brasiliensis* (von Ihering), *Leptopilina* sp., *Hexacola* sp. [40], parasitism levels were extremely low. The figitid specimens recovered from *D. suzukii* were taxonomically similar to *L. boulardi*, a worldwide saprophytic drosophilid’s parasitoid. *Leptopilina boulardi* was recently associated with *D. suzukii* in Argentina (Vanina Reche, unplublished data). In contrast to the two invasive fly species, five of the six identified parasitoid species (83%) were recovered from the native *A. fraterculus*. In addition, the highest parasitoid abundance in *P. guajava* and *J. australis* came from *A. fraterculus*. *Anastrepha fraterculus* was mostly parasitized by *G. pelleranoi*, followed by two native braconid parasitoids, *D. areolatus* and *D. brasiliensis*, whereas sporadically by the pupal parasitoids *P. vindemiae* and *T. anastrephae*. Both *Doryctobracon* species integrate an assemblage of several Neotropical-native parasitoids that co-evolved in sympatry with *A. fraterculus* in South American rainforest areas [76].

## 5. Conclusions

The current study improves our understanding of the temporal and spatial dynamics of these three important pest fruit flies, the utilization patterns and relative importance of non-crop hosts for these pests, as well as the trophic associations with resident parasitoids in the disturbed non-crop habitats surrounding cultivated crops. As shown in this study, the disturbed natural habitat would inevitably provide sources of the fly populations that may move into adjacent fruit crops. The three pests also showed different host preferences. Both *C. capitata* and *D. suzukii* preferred peach and loquat, their highest infestation levels thus occurred between December and February when peaches were highly available. In contrast, high levels of infestations by *A. fraterculus* occurred between February and April when guavas were highly available. Both *P. vindemiae* and *T. anastrephae* are key natural mortality factors of *D. suzukii* while *G. pelleranoi* is the main natural mortality factor of both *C. capitata* and *A. fraterculus*. Consequently, area-wide management strategies must consider reducing pest pressure in susceptible crops by reducing sources of fly populations in the non-crop habitats. In this context, biological control is highly desirable to naturally regulate the fly populations. The current study suggests that timed mass releases of these parasitoids during early or peak infestation stages of these pests in disturbed habitats may help suppress the fly populations prior to their main spread to commercial crops. 

## Figures and Tables

**Figure 1 insects-15-00061-f001:**
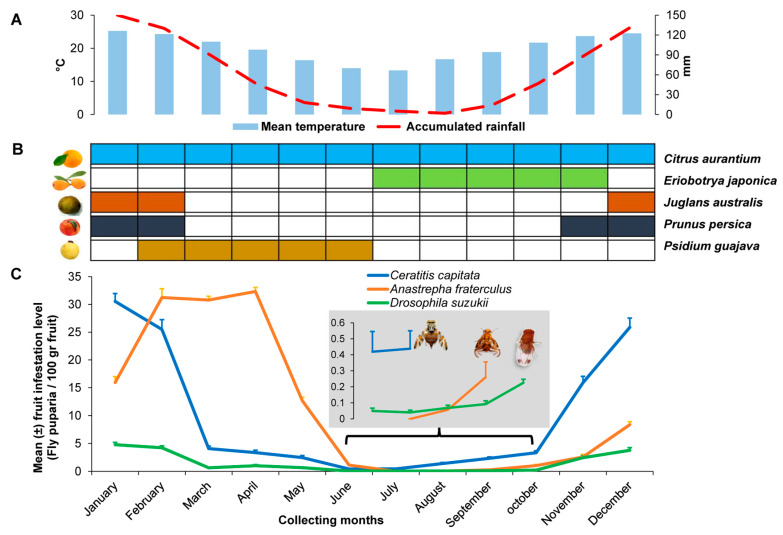
Mean temperature and accumulated rainfall during collecting periods (**A**), temporal patterns of availability for five host plant species (**B**), and seasonal dynamics of total fruit infestation levels (data were pooled from different host plants) of *Ceratitis capitata*, *Anastrepha fraterculus*, and *Drosophila suzukii* (**C**) at the study site (Horco Molle, Tucuman, northwestern Argentina).

**Figure 2 insects-15-00061-f002:**
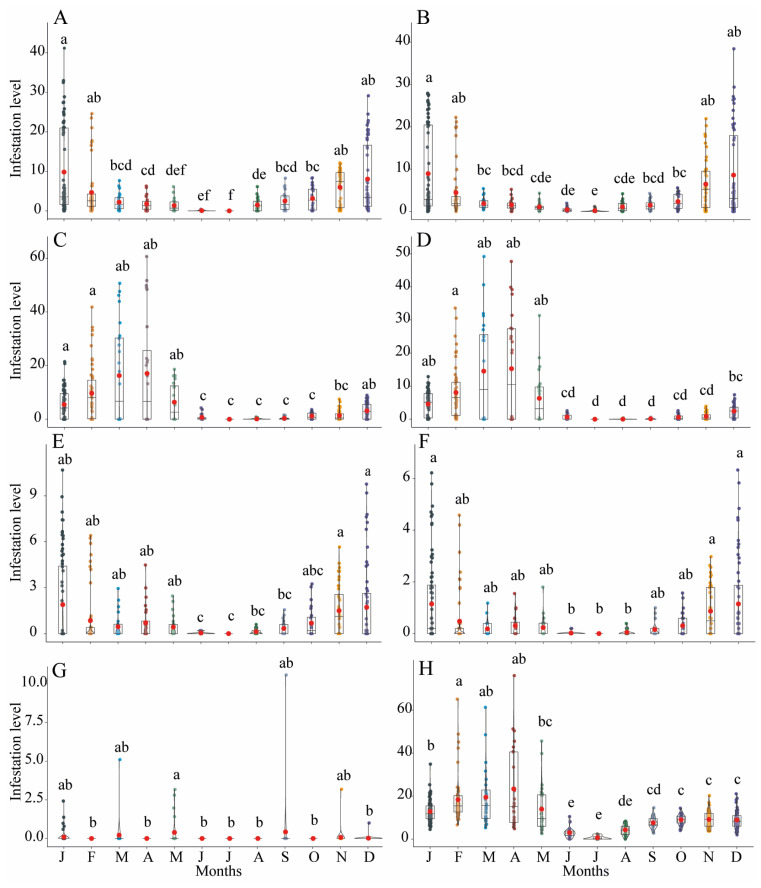
Monthly fruit infestation levels (fly puparia/100 g fruit) by *Ceratitis capitata* (**A**,**B**), *Anastrepha fraterculus* (**C**,**D**), *Drosophila suzukii* (**E**,**F**), and *Drosophila* spp. (*D. melanogaster* group) (**G**,**H**) recorded from fruits collected from tree canopies (left column) and from the ground (right column) at the study site (Horco Molle, Tucuman, northwestern Argentina). Different lowercase letters represent significant differences at α = 0.05 (Dunn’s Test).

**Figure 3 insects-15-00061-f003:**
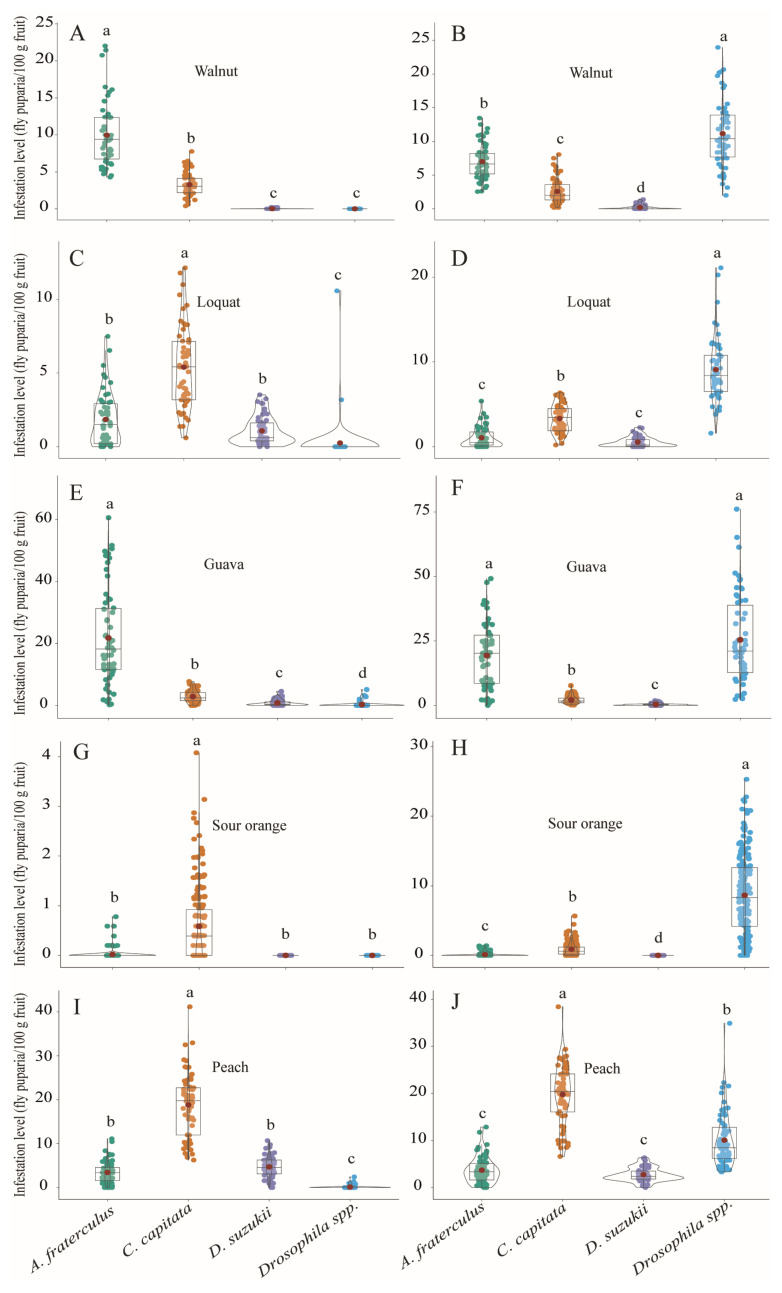
Infestation levels (fly puparia/100 g fruit) of *Anastrepha fraterculus*, *Ceratitis capitata*, *Drosophila suzukii*, and *Drosophila* spp. (*D. melanogaster* group) recorded from host fruit species collected from tree canopies (**left column**) and from the ground (**right column**) in Walnut (**A**,**B**), loquat (**C**,**D**), guava (**E**,**F**), sour orange (**G**,**H**), and peach (**I**,**J**) at the study site (Horco Molle, Tucuman, northwestern Argentina). Different lowercase letters represent significant differences at α = 0.05 (Dunn’s Test).

**Figure 4 insects-15-00061-f004:**
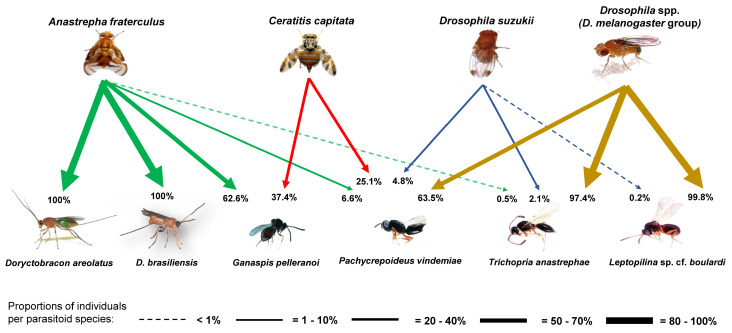
The proportion of individuals belonging to a particular parasitoid species over the total number of specimens from that parasitoid species recovered from each host fly species during collection periods at the study site (Horco Molle, Tucuman, northwestern Argentina).

**Figure 5 insects-15-00061-f005:**
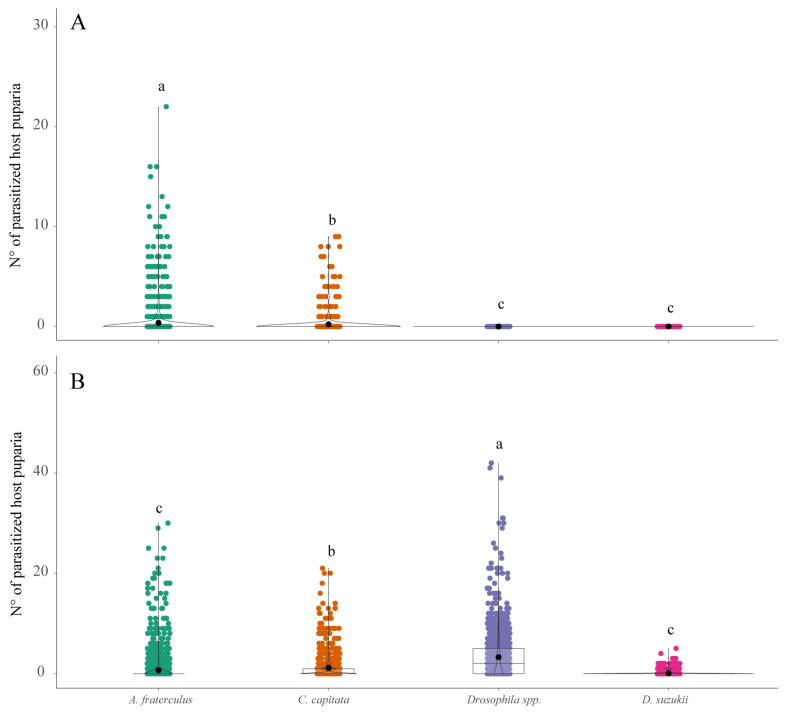
Comparison of the parasitized puparia number of *Anastrepha fraterculus*, *Ceratitis capitata*, *Drosophila* spp. (*D. melanogaster* group), and *Drosophila suzukii* recovered from (**A**) fallen fruits on the ground and (**B**) fruits still on the canopies at the study site (Horco Molle, Tucuman, northwestern Argentina). Different lowercase letters represent significant differences at α = 0.05 (Dunn’s Test).

**Figure 6 insects-15-00061-f006:**
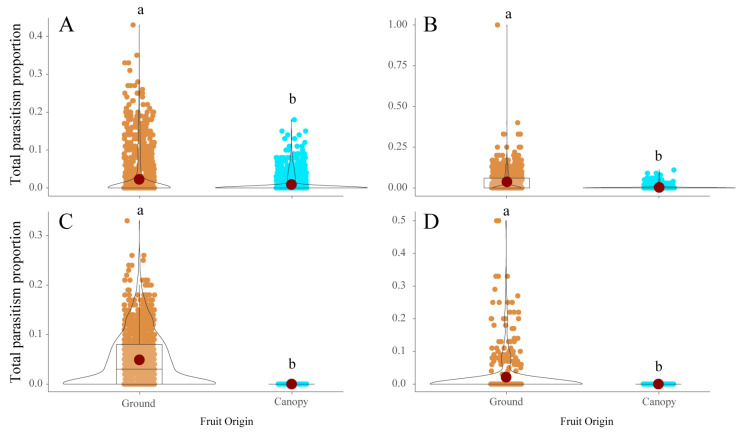
Total parasitism proportion recorded on (**A**) *Anastrepha fraterculus*, (**B**) *Ceratitis capitata*, (**C**) *Drosophila* spp. (*D. melanogaster* group), and (**D**) *Drosophila suzukii* from fruit samples collected from both ground and tree canopies at the study site (Horco Molle, Tucuman, northwestern Argentina). Different lowercase letters represent significant differences at α = 0.05 (Mann-Whitney’s test).

**Table 1 insects-15-00061-t001:** Total numbers of *Anastrepha fraterculus* (*Af*), *Ceratitis capitata* (*Cc*), *Drosophila suzukii* (*Ds*), and *Drosophila* spp. from *D. melanogaster* group (*D*spp) *puparia*, and emerged adult flies, recovered from *Citrus aurantium* (*Ca*), *Eriobotrya japonica* (*Ej*), *Juglans australis* (*Ja*), *Prunus persica* (*Pp*), and *Psidium guajava* (*Pg*) fruits collected from canopies and ground between November 2016 and March 2020 in Horco Molle, Tucumán, northwestern Argentina.

Fruit Origin	Fruit Species	No. of Collected Fruit (Weight, Kg)	Total Numbers							
			*Af* Puparia	*Af* Adults	*Cc* Puparia	*Cc* Adults	*Ds* Puparia	*Ds* Adults	*D*spp Puparia	*D*spp Adults
**Canopy**	*Ca*	692 (87.4)	17	8	514	203	0	0	0	0
	*Ej*	2700 (26.9)	492	245	1442	763	286	144	16	11
	*Ja*	672 (28.3)	2819	1437	923	493	4	2	0	0
	*Pp*	960 (32.6)	1122	550	6120	2948	1537	725	36	23
	*Pg*	580 (29.1)	6321	3059	824	358	224	86	73	47
**Ground**	*Ca*	92 (87.8)	108	45	767	301	0	0	7595	3158
	*Ej*	2700 (27.1)	291	94	895	336	148	41	2458	1015
	*Ja*	672 (28.1)	1974	832	724	299	61	23	3145	1209
	*Pp*	960 (32.3)	1195	543	6376	3009	887	374	3249	1339
	*Pg*	580 (29.2)	5650	2350	612	228	95	25	7427	2857

**Table 2 insects-15-00061-t002:** Summary of Kruskal–Wallis models on the infestation levels by *Ceratitis capitata*, *Anastrepha fraterculus*, *Drosophila suzukii*, and *Drosophila* spp. (*D. melanogaster* species group) on *Citrus aurantium*, *Eriobotrya japonica*, *Juglans australis*, *Prunus persica*, and *Psidium guajava* fruits collected from both canopies and the ground during fruiting seasons between November 2016 and March 2020 in Horco Molle, Tucumán, Northwestern Argentina.

Fruit Origin: Fruit Species	Statistical Results			
	*df*	*n*	χ^2^	*p*
**Canopy:**				
*Citrus aurantium*	3	348	316.90	<0.0001
*Eriobotrya japonica*	3	108	145.38	<0.0001
*Juglans australis*	3	112	209.24	<0.0001
*Prunus persica*	3	128	208.25	<0.0001
*Psidium guajava*	3	116	164.67	<0.0001
**Ground:**				
*Citrus aurantium*	3	348	485.34	<0.0001
*Eriobotrya japonica*	3	108	155.39	<0.0001
*Juglans australis*	3	112	179.20	<0.0001
*Prunus persica*	3	128	181.80	<0.0001
*Psidium guajava*	3	116	173.21	<0.0001

## Data Availability

The dataset is available at the CONICET-Digital Institutional Repository, URI: http://hdl.handle.net/11336/221026, accessed on 8 January 2024.

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
