# Peer review of "The Population Dynamics and Parasitism Rates of Ceratitis capitata, Anastrepha fraterculus, and Drosophila suzukii in Non-Crop Hosts: Implications for the Management of Pest Fruit Flies"

_insects, 2024, doi:10.3390/insects15010061_

Round 1
Reviewer 1 Report
Comments and Suggestions for Authors
The manuscript entitled Population Dynamics and Parasitism Rates of Ceratitis capitata, Anastrepha fraterculus, and Drosophila suzukii in Non-Crop Hosts: Implications for Area-wide Management of Pest Fruit Flies presents very interesting and significant biological information needed to develop integrated fruit fly pest management strategies that minimize environmental impact and maximize long-term sustainability.
The topic is of relevance not only for Argentina but for the rest of South America and also other fruit growing areas in which Drosophila suzukii has become a serious problem along with other invasive species. The experimental procedures and data analysis is correct. The manuscript is well written and in a sufficient level of detail. The discussion is in accordance with the results obtained. The results provided and its interpretation are valuable for the improvement of fruit fly integrated pest management. Further, results can be extrapolated to other areas.
I have a few minor comments which I have included as notes in the manuscript.
In all the manuscript is worth being published. I congratulate the authors for such interesting and complete study. I hope my minor comments contribute to a better visualization of it.

Author Response
Responses to Reviewer 1' comments (highlighted):
The manuscript entitled Population Dynamics and Parasitism Rates of Ceratitis capitata, Anastrepha fraterculus, and Drosophila suzukii in Non-Crop Hosts: Implications for Area-wide Management of Pest Fruit Flies presents very interesting and significant biological information needed to develop integrated fruit fly pest management strategies that minimize environmental impact and maximize long-term sustainability.
The topic is of relevance not only for Argentina but for the rest of South America and also other fruit growing areas in which Drosophila suzukii has become a serious problem along with other invasive species. The experimental procedures and data analysis is correct. The manuscript is well written and in a sufficient level of detail. The discussion is in accordance with the results obtained. The results provided and its interpretation are valuable for the improvement of fruit fly integrated pest management. Further, results can be extrapolated to other areas.
I have a few minor comments which I have included as notes in the manuscript. In all the manuscript is worth being published. I congratulate the authors for such interesting and complete study. I hope my minor comments contribute to a better visualization of it.
We thank the reviewer for the valuable comments and corrections directly on the PDF version which helped improving the manuscript and have clarified any points of confusion. All suggested changes as marked on the PDF version have been accepted and incorporated into the revision.
As recommended by the reviewer, we have now merged Figs. 2 and 3. Please note that we've removed the old figures before inserting the new ones.
Reviewer 2 Report
Comments and Suggestions for Authors
There is a need of improving knowledge of the interaction between invasive and native fruit fly species, their host plants, and their parasitoids in environments that are disturbed but in the process of restoration. The current study aims to describe the abundance of and infestation levels by two invasive fruit fly species (C. capitata and D. suzukii) and A. fraterculus, a native species, over a 40-month period. The article is well written. I believe that the results add novel angles to this body of knowledge, and therefore I recommend publication in Insects after a revision is made. Such a revision can be seen aa major or minor depending on the author’s perspective.
The length of the article is acceptable, except for the Conclusions. This section needs to be cut at least in half. The authors tried to summarize their main findings, but a better job needs to be done.
Two concerns I have are (1) what is the relevance of reporting saprophytic drosophilids? I didn’t see a need to report those results, which are just a result of having collected fruits from the ground, and (2) the authors indicate in the title that their results have implications for area-wide IPM. That word ‘area-wide’ is found once in the title, and twice in the Introduction. After that, it was not discussed at all.
I suggest the authors try to make the Discussion a bit more concise. In general, that section is okay but sometimes it is hard to connect their findings with actual implications for management. Are insecticides being used in cropped areas near non-cropped areas? I feel that the fruit fly management part is somewhat missing.
Attached is a PDF file that includes numerous minor edits that need to be made. All figures need to be improved quality-wise.
Overall, a good article that just needs some polishing.

Minor yet numerous edits are needed. See PDF file.
Author Response
Responses to Reviewer 2's comments (highlighted):
There is a need of improving knowledge of the interaction between invasive and native fruit fly species, their host plants, and their parasitoids in environments that are disturbed but in the process of restoration. The current study aims to describe the abundance of and infestation levels by two invasive fruit fly species (C. capitata and D. suzukii) and A. fraterculus, a native species, over a 40-month period. The article is well written. I believe that the results add novel angles to this body of knowledge, and therefore I recommend publication in Insects after a revision is made. Such a revision can be seen a major or minor depending on the author’s perspective.
The length of the article is acceptable, except for the Conclusions. This section needs to be cut at least in half. The authors tried to summarize their main findings, but a better job needs to be done.
We thank the reviewer for the valuable comments and corrections which helped improving the manuscript and have clarified any points of confusion. All suggested changes as marked on the PDF version have been accepted and incorporated into the revision.
We agree that the original conclusion is too long. As suggested by the reviewer, we have re-written the conclusion and substantially reduced the length of this section from originally 469 words to 232 words.
Two concerns I have are (1) what is the relevance of reporting saprophytic drosophilids? I didn’t see a need to report those results, which are just a result of having collected fruits from the ground, and (2) the authors indicate in the title that their results have implications for area-wide IPM. That word ‘area-wide’ is found once in the title, and twice in the Introduction. After that, it was not discussed at all.
Although most of these saprophytic drosophilids are not considered as pests, they are common hosts for those generalist drosophila parasitoids, and may share these generalist parasitoids species with the pest D. suzukii or act as alternative hosts when D. suzukii is not available. It is thus important to document the occurrences and population dynamics as well as associated parasitoid species of these saprophytic drosophilids in this study. We now added brief discussion on the relevance of saprophytic drosophilids in this study in the introduction and discussion sections.
We’ve removed the “area-wide” from the title, as we realized that this study was focused mainly on the population dynamics of these pest fruit flies and their associated parasitoids in a non-crop habitat. However, to develop area-wide programs for these polyphagous and highly mobile pests, it is critical to understand how these pest populations persist and disperse in the landscape as the season progresses. We have now re-written the conclusions to emphasis the potential implications of this study for area-wide management of these pests.
I suggest the authors try to make the Discussion a bit more concise. In general, that section is okay but sometimes it is hard to connect their findings with actual implications for management. Are insecticides being used in cropped areas near non-cropped areas? I feel that the fruit fly management part is somewhat missing.
We agree and have tried to make the discussion section as concise as possible. We did not monitor /document the management practices of the adjacent crops. It is possible that these crops might have been sprayed with pesticides. The potential implication of this study for the management of these pest fruit fly is now strongly stated in the conclusions.
Attached is a PDF file that includes numerous minor edits that need to be made. All figures need to be improved quality-wise.
We appreciate all edits which are very helpful. All edits have been incorporated into the revision. Unfortunately, when these figures were inserted into the journal’s template they had to be reduced in size and when the manuscript was converted into the PDF, somehow these figures have lost the original resolutions and look blurry. We have now remade all figures and reduced the original sizes of these figures (that can fit into the template directly) and they now look quite clear. Please note that all old figures have been removed before we inserted the new ones.
Overall, a good article that just needs some polishing.
Thanks for the positive response.